# Improvement of Yttrium Oxyfluoride Coating with Modified Precursor Solution for Laser-Induced Hydrothermal Synthesis

**Jaeho Park** [1,2], **Kyungwoo Lee** [1], **Jaehong Lee** [1], **Hae Won Hwang** [1,2], **Goeen Jeong** [1,2], **Kyung Yeun Kim** [1], **Yu-Chan Kim** [1], **Myoung-Ryul Ok** [1], **Hyung-Seop Han** [1], **Jeong-Yun Sun** [2,*] and **Hojeong Jeon** [1,3,*]

[1] Biomaterials Research Center, Biomedical Research Division, Korea Institute of Science and Technology (KIST), Seoul 02792, Korea; haha0728@kist.re.kr (J.P.); t21063@kist.re.kr (K.L.); jaehong1722@kist.re.kr (J.L.); haehwang@kist.re.kr (H.W.H.); jeong056@kist.re.kr (G.J.); ky.kim@kist.re.kr (K.Y.K.); chany@kist.re.kr (Y.-C.K.); omr2da@kist.re.kr (M.-R.O.); hyuhan@kist.re.kr (H.-S.H.)
[2] Department of Materials Science and Engineering, Seoul National University (SNU), Seoul 08826, Korea
[3] KU-KIST Graduate School of Converging Science and Technology, Korea University, Seoul 02841, Korea
[*] Correspondence: jysun@snu.ac.kr (J.-Y.S.); jeonhj@kist.re.kr (H.J.)

**Abstract:** In the semiconductor manufacturing process, the inner walls of the equipment are coated with yttrium-based oxides for etch resistance against plasma exposure. Yttrium oxyfluoride (YOF) particle synthesis and coating methods have been actively studied owing to their high erosion resistance compared to $Y_2O_3$ and $Al_2O_3$. Owing to the formation of a rough and porous coating layer by thermal spray-coating, the coating layer disintegrates, as the etching process has been conducted for a long time. Laser-induced synthesis and coating technology offer several advantages, including simplified process steps, ease of handling, and formation of a dense coating layer on the target material. In this study, YOF was coated on an aluminum substrate using a modified precursor solution. The NaF and HMTA were added to the precursor solution, resulting in enhanced synthetic reactivity and stabilizing the oxides. The material coated on the surface was analyzed based on the characteristics of composition, chemical bonding, and phase identification. We found that the coating properties can be improved by using an appropriate combination of modified precursor solutions and laser parameters. Therefore, the findings in this study are expected to be utilized in the field of coating technology.

**Keywords:** laser-induced single-step coating; yttrium oxyfluoride (YOF); laser-induced hydrothermal reaction

## 1. Introduction

A plasma-etching process that can fabricate a sophisticated linewidth is essential for producing highly integrated products in the semiconductor industry [1–4]. This etching process proceeds in a fluorinated hydrocarbon plasma environment, and reacts with various ceramic components in the chamber, resulting in the erosion of parts and generation of contaminated particles. The byproduct particles drop on the wafer and induce a flaw, resulting in a significant yield loss. Therefore, it is essential to cover the inner walls of the chamber with a ceramic substance that has high hardness, wear resistance, dielectric strength, corrosion resistance, and chemical stability.

Among the plasma-resistant stable materials, yttrium oxide ($Y_2O_3$) is extensively employed as a plasma-facing wall-coating material, because its etching rate is significantly lower than that of silicon dioxide ($SiO_2$) and aluminum oxide ($Al_2O_3$). However, $Y_2O_3$ is degraded when interacting with corrosive gases, such as fluorocarbides—including $CF_4$, $CHF_3$, $C_4F_6$, and $C_2F_6$—disassembling the Y-O bond structure. Therefore, substances with better anti-plasma properties than $Y_2O_3$ are being developed [5–7].

Yttrium oxyfluoride (YOF) has attracted attention as an alternative to $Y_2O_3$ owing to its improved etching resistance [8–15]. YOF is a compound produced by the reaction

of $Y_2O_3$ with fluorine plasma, and is more chemically stable than $Y_2O_3$. The enthalpy of formation energy for the metal–oxygen bonding of YOF is $-392$ kJ mol$^{-1}$, which is less than the $Y_2O_3$ value of $-318$ kJ mol$^{-1}$ [16,17]. Therefore, research is being actively conducted on the synthesis and effective coating of YOF.

Thermal spray-coating technology, utilizing high-temperature thermal energy as a source, has been used to coat such oxide materials. In this procedure, powder composed of yttrium is applied as a coating material source for thermal spray-coating. However, depending on the powder size, the roughness and porosity of the coating layer increase, causing the coating layer to disintegrate as the etching process proceeds [18–21]. To address this issue, several studies have suggested developing a post-treatment technique to improve the binding ability and density of the coating layer [22,23].

Recently, our group developed a technology for synthesizing and coating ceramic materials using laser heat treatment [24]. The laser-induced single-step coating (LISSC) technology offers numerous advantages, because it allows coating at the same time as particle synthesis [25–27]. The LISSC process is comparable to hydrothermal synthesis, which uses heat as the source of energy. The preparation of the process is straightforward: the substance in which the coating material is to be deposited and the precursor solution containing the coating material are determined. Furthermore, it is feasible to apply a selective coating via a local focal region of a laser beam, or a large-area coating by laser scanning on the surface. The coating process variables can be adjusted without pretreatment or vacuum processes, resulting in ease and cost-effectiveness. Various types of oxides can be deposited, depending on the type of precursor [28–32].

In this study, a YOF ceramic was synthesized and coated on aluminum, which was used as the inner side of the plasma-etching chamber via LISSC processing. Because Al has high thermal conductivity (237 W/(m K)), it reacts with precursor ions and laser energy on the surface relatively quickly. To address this issue, the reaction on the surface increased the stability of the synthesis by modifying the precursor components. Through a series of experiments, surface morphological, chemical, and crystallographic measurements were performed to determine the optimal deposition conditions.

## 2. Materials and Methods

### 2.1. Preparation of Coated Substrates and Precursor Solutions

A plate of pure Al of 10 mm $\times$ 10 mm $\times$ 1 mm in size was prepared and polished with 1000-grit sandpaper to remove debris and form a smooth surface. The polished samples were sonicated (SD-3000H, Mujigae, Seoul, Korea) in 99% ethanol for 10 min, and then dried in the air. The washed sample was fixed in a Petri dish with a diameter of 60 mm and immersed in an yttrium precursor solution before laser treatment (Figure 1).

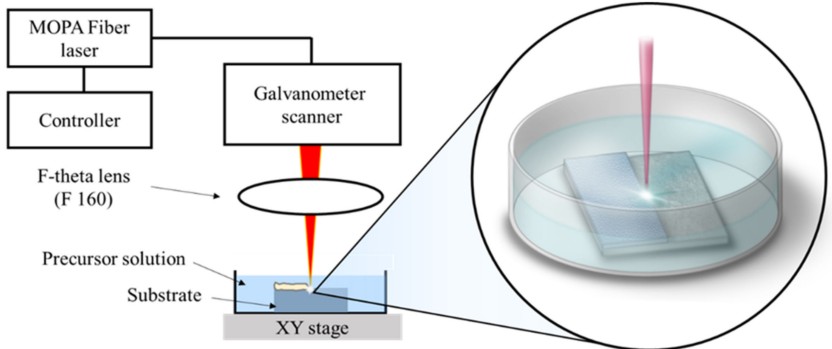

**Figure 1.** Illustration of laser-induced single-step coating (LISSC) in the precursor solution.

A 1 M solution of yttrium nitrate hexahydrate (Y(NO$_3$)$_3$ 6H$_2$O; 237957, Sigma-Aldrich, St. Louis, MS, USA) was used to supply a sufficient amount of yttrium. Then 1 M of sodium fluoride (NaF; S6776, Sigma-Aldrich, St. Louis, MS, USA) for providing fluorine (F), and

0.5 M of hexamethylenetetramine (HMTA; 398160, Sigma-Aldrich, St. Louis, MS, USA) for hydroxyl group stabilization were mixed to prepare a modified precursor solution. To minimize the reaction in the solution before laser-induced synthesis processing, each compound was dissolved in distilled water and sequentially added in the following order: $Y(NO_3)_3$, NaF, and HMTA. NaF was added to supply fluorine to the YOF coating formation, and HMTA was further added to form a more stable oxide [19,33–35]. The precursor solution mixed with all chemicals was selected for the evaluation of YOF synthesis. The total combined dose was dissolved to 10 mL. Before laser synthesis, the mixed precursor solution remained transparent, without particle formation.

### 2.2. Laser-Induced Single-Step Coating (LISSC) Process

The equipment used for coating was a nanosecond (ns) ytterbium fiber laser (IPG Laser, Burbach, Germany) with a wavelength of 1064 nm. The laser beam size was 60 μm, and emitted a Gaussian $TEM_{00}$ mode. All samples were scanned at a rate of 500 mm/s using a Galvo scanner (SCANcube 10, Scanlab, Puchheim, Germany) with a repetition rate of 100 kHz. The pulse width of the beam was 200 ns, and three average laser powers of 5 W, 10 W, and 18.3 W were used in this process. Laser fluence was calculated by dividing the average power by the beam area. The spacing distance between the laser beamlines was fixed at 50 μm, resulting in laser treatment of the entire area of the samples with a zigzag pattern. To optimize the surface coating conditions, the number of loops (L) was scanned 10, 50, 100, 150, and 200 times (Table 1). The L-10, 50, 100, 150, and 200 indicate the number of times that the laser scan was performed in the same treated region. A detailed schematic of laser-induced single-step coating (LISSC) is shown in Figure 1.

**Table 1.** Laser processing parameters.

| Laser Parameter | Value, Property |
| --- | --- |
| Wavelength [nm] | 1064 |
| Laser power [W] | 5, 10, 18.3 |
| Pulse width [ns] | 200 |
| Repetition rate [kHz] | 100 |
| Laser fluence [W/cm$^2$] | $1.77 \times 10^5$, $3.53 \times 10^5$, $6.47 \times 10^5$ |
| Scan speed [mm/s] | 500 |
| Spot diameter [μm] | 60 |
| Focal length [mm] | 160 |

### 2.3. Surface Characterization

The surface morphology and components were analyzed via scanning electron microscopy (SEM; Inspect F50, FEI, Hillsboro, OR, USA) equipped with energy-dispersive spectroscopy (EDS). The degree of YOF coating was estimated to be the degree to which yttrium (Y) was included in the coating layer. Since the amount of yttrium determines the formation of the YOF coating layer indirectly, optimization experiments were conducted to increase the content of yttrium after laser treatment. It was confirmed that the quantity of Y was changed according to each solution condition and laser parameter. The exact composition and characteristics of coating layer were analyzed through a series of measurements. The surface roughness of the samples was analyzed using a confocal laser scanning microscope (LSM; LEXT OLS4100, Olympus, Tokyo, Japan). To measure the surface wettability change, contact-angle measurements (SmartDrop, Femtobiomed, Gyeonggi-do, Korea) were performed by adding 3 μL of deionized water. The YOF-coated surface was scanned by X-ray diffraction (XRD; D8 ADVANCE, Bruker, Madison, WI, USA) in thin-film mode using a Cu Kα radiation beam for phase analysis before and after laser coating. The wavelength of 1.54 Å radiation with 40 kV and 40 mA was used, and the incident angle of radiation was 1°. Owing to the thin-film-mode XRD, drastically different XRD peak data could be obtained. Data were collected from 10° to 90° at 0.02° step intervals. Chemical bonding on the YOF surface was investigated by X-ray photoelectron spectroscopy (XPS;

PHI 5000, ULVAC-PHI, Chigasaki, Japan). The core states of Y, O, and F were scanned using a monochromatized Al K$\alpha$ line. The beam size was $100 \times 100$ $\mu m^2$ under a base pressure of $2.0 \times 10^{-7}$ Pa.

## 3. Results

### 3.1. Optimization for Stable YOF Formation

We first examined the conditions under which yttrium was coated on the Al surface.

Figure 2 shows SEM images after laser treatment of the yttrium nitrate solution. At the power of 5 W and 10 W, the formation of a coating layer was not observed even when the number of loops was increased. Moreover, at the power of 18.3 W, the substrate was melted due to the high laser beam energy, forming a morphology with high porosity and roughness. The formation of the coating layer did not occur under any conditions, but the surface morphology was changed by increasing roughness. Surfaces with high porosity and roughness at the power of 18.3 W were considered unsuitable conditions because the increased surface area reacted with plasma gas in a corrosive environment.

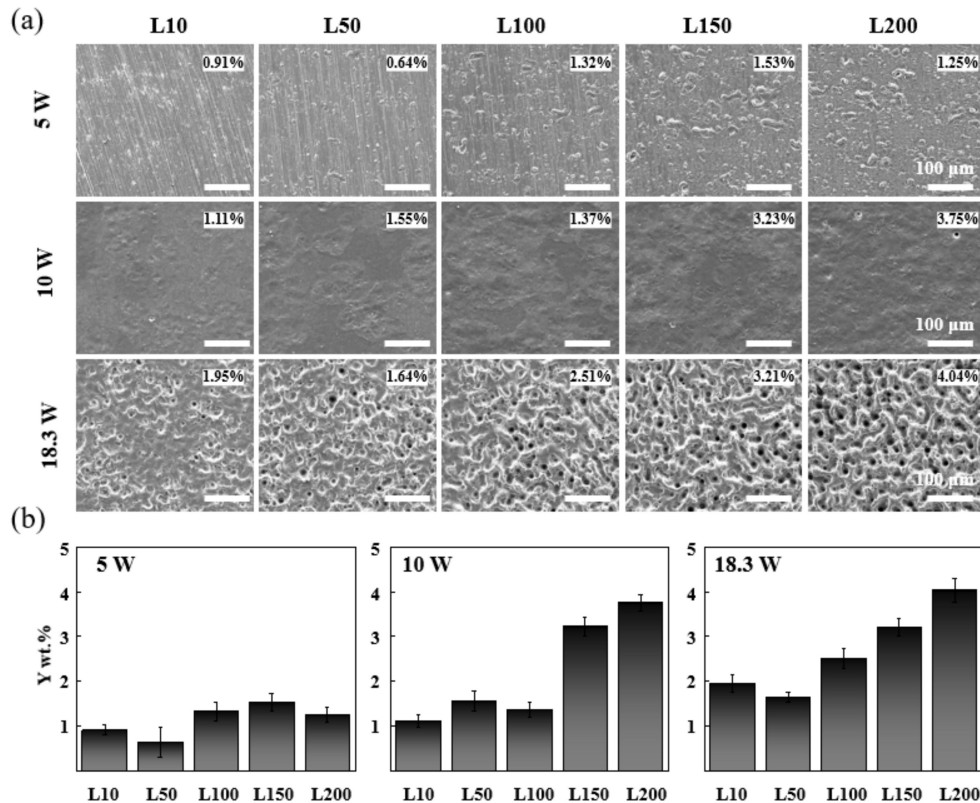

**Figure 2.** (**a**) SEM images of laser-treated Al showing surface morphology, and (**b**) weight percentage of coated Y on the Al substrate depending on laser parameters (power, loop) in the Y(NO$_3$)$_3$ precursor solution. In the SEM images, the numerical values indicate the weight percentage of yttrium.

Following EDS elemental analysis, the degree of formation of the coating layer was confirmed by the change in the yttrium composition. When the surface was irradiated at a power of 5 W, the surface morphology changed as the number of loops increased, but the amount of the Y component on the surface was barely measured at 5 wt % (Figure 2b). For the surface irradiated with an increasing power of 18.3 W, melting and ablated surface areas were observed owing to the high fluence of the laser beam energy. With a diameter of 60 $\mu m$ at the focused spot of the laser beam, the energy output of 18.3 W indicates a laser fluence of $6.47 \times 10^5$ W/cm$^2$, which has sufficient energy to melt the metal surface [36]. Although the content of Y slightly increased compared to the lower powers at 5 W and

10 W, it was necessary to modify the conditions because of the rough surface and low Y content.

Next, the conditions under which YOF was coated on the aluminum surface were examined using a precursor solution containing $Y(NO_3)_3$ and NaF. Figure 3 shows the laser-induced coated surface in the precursor solution containing fluorine, which is a component of YOF and a strong oxidizing agent [37,38]. The polishing trace diminished as the loop increased at the power of 5 W and 10 W. It was confirmed that a compound was synthesized on the surface, demonstrating that a reaction was facilitated after NaF was mixed with the precursor solution. On the other hand, at the power of 18.3 W, a porous surface was observed without the formation of a coating surface. Even when the loop and power were increased, there was no significant change in morphology above a certain level. Following the addition of NaF to the precursor solution, the quantity of Y on the surface was significantly increased above L50 in the range of all laser power conditions (Figure 3b). In addition, a laser-treated surface with a smooth surface was obtained owing to the formation of an oxide layer supported by the Y precursor solution containing NaF. However, even under these adjusted solution conditions, the formation of the laser-induced coating was hampered at a high laser power of 18.3 W, owing to the effects of laser melting and ablation.

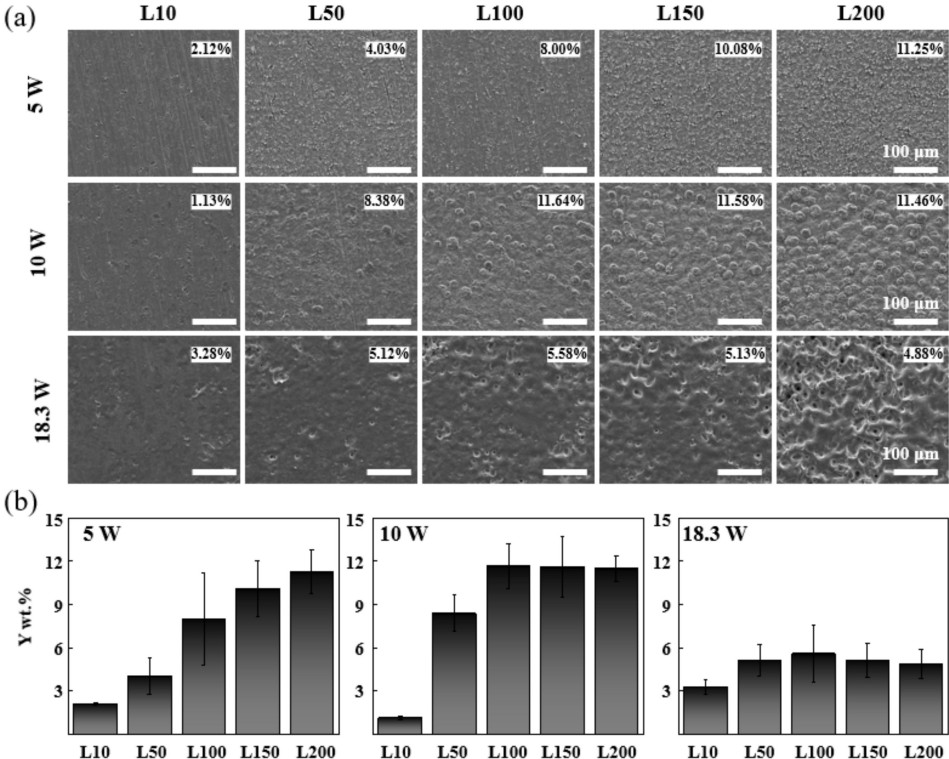

**Figure 3.** (**a**) SEM images of the laser-treated Al substrate showing surface morphology, and (**b**) weight percentage of coated Y on the Al substrate depending on laser parameters (power, loop) in a mixture of $Y(NO_3)_3$ and NaF precursor solution. In the SEM images, the numerical values indicate the weight percentage of yttrium.

To enhance the formation of YOF on Al substrates, HMTA, which induces hydroxyl group stabilization, was added to the precursor solution [39–41]. The surface was analyzed after laser-induced coating in an yttrium nitrate solution containing NaF and HMTA (Figure 4). As the number of laser treatments increased (L increased), an even coating surface was formed without pores at a power of 5 W. At the power of 10 W and 18.3 W, the synthetic material was coated, but the surface was formed with pores and roughness. EDS component analysis was used to check the level of coating formation. A larger quantity of

Y was detected, owing to the action of stabilizing the oxide agent—up to approximately 30 times more than that of the unmodified nitrate solution. The Y content showed a tendency to saturate from L100 at a power of 5 W (Figure 4b). However, as the power increased to 5 W and 10 W, more traces of the melted surface appeared, rather than the precipitation and coated YOF. In addition, because the synthesis reactivity was increased by the influence of the additional chemical, the higher the power, the more the material synthesized in solution was dispersed as particles rather than being coated on the substrate. However, the coating was not achieved in the power range of less than 5 W, because the supplied energy was insufficient for the synthesis reaction. Consequently, additional analyses were performed using the optimal laser-coating conditions based on a power of 5 W and a loop of 100.

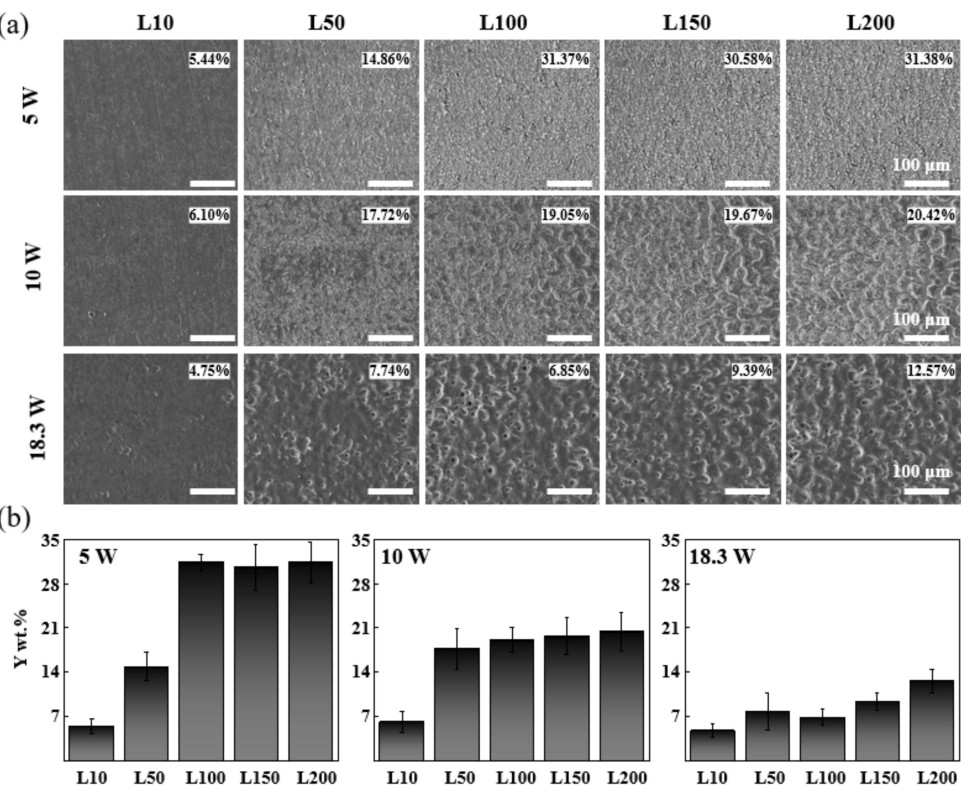

**Figure 4.** (**a**) SEM images of the laser-treated Al substrate showing surface morphology, and (**b**) weight percentage of coated Y on the Al substrate depending on laser parameters (power, loop) in a mixture of $Y(NO_3)_3$, NaF, and HMTA precursor solution. In the SEM images, the numerical values indicate the weight percentage of yttrium.

As shown in Figure 5b, even after a series of laser treatments, the formation of the coating layer was non-uniform. The gray area (indicated by arrows) had a relatively high Y content compared to other areas; however, only a content within approximately 7 wt % was observed. The lack of a homogeneous coating layer was assumed to be due to the insufficient time on the surface of the coating material to be synthesized, because of the high heat conductivity of aluminum. However, when an additional chemical was added, the laser coating was uniformly applied to the substrate (Figure 5c,d). The weight percentage of Y at each surface condition was investigated using EDS for quantitative analysis, as shown in Figure 5e. After laser treatment under each precursor solution condition, the quantities of Y were evaluated as 3.6%, 10.93%, and 31.11%. The final modified precursor solution showed a 10-fold higher Y content than that of the $Y(NO_3)_3$ precursor solution. In this study, the modified solution contained HMTA, which was used for hydroxyl stabilization, resulting in a complementary role in the formation and stable deposition of YOF. It was confirmed that the amount of YOF on the surface increased as the reactivity of the synthesis

increased, resulting in an improvement in the coating ability of the deposited synthesized YOF. Therefore, an even surface with a higher Y content was formed by the combination of NaF and HMTA, which enhanced the synthetic reactivity of YOF. Thus, an effective coating is possible depending on the appropriate combination of the modified precursor solutions and laser parameters.

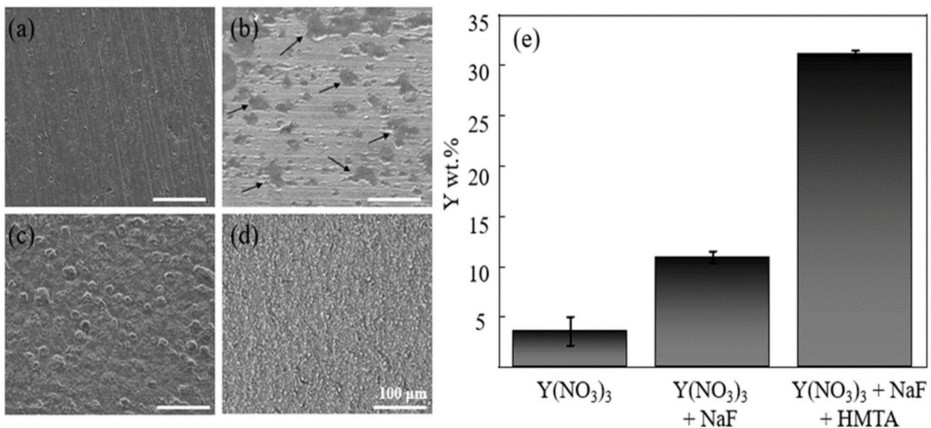

**Figure 5.** SEM images of the Al substrate (**a**) before and after the LISSC process in (**b**) $Y(NO_3)_3$, (**c**) a mixture of $Y(NO_3)_3$ and NaF, and (**d**) a mixture of $Y(NO_3)_3$, NaF, and HMTA solution. (**e**) The quantitative EDS data for the weight percentage of Y component distribution on Al after the coating process, by precursor solution type.

### 3.2. Component Analysis of Coated Surfaces

A series of analyses were performed to confirm the components and surface properties of the coated YOF layer. The surface before and after laser-induced coating using a modified solution is shown in Figure 6a. The surface roughness (Sa) increased slightly after the laser treatment, and was measured at 0.214 μm (bare substrate) and 0.935 μm (coated surface). The roughness change could be minor compared to the spray-coating method employing microscale powder as a coating source. In addition, the thickness of the coating layer was observed to be within 1 μm without an interface between the coating and substrate (Figure 6b). This result was considered to be strongly bonded to the surface owing to the gradient of the laser heat energy. The surface properties after the laser treatment were evaluated to demonstrate the performance of the coating layer. The contact angle was used to compare the surface characteristics before and after the coating treatment (Figure 6c). Depending on the surface properties, the water-contact angle sharply increased from 77.8° (bare substrate) to 159.3° (coated surface). Because YOF is a stable chemical-like oxide, the surface energy decreased in the YOF-coated area, resulting in superhydrophobic wetting. Figure 6d shows a mapping image of the YOF-coated surface using the modified precursor solution, including $Y(NO_3)_3$, NaF, and HMTA. Each component element was uniformly distributed without agglomeration on the Al surface. Therefore, it was confirmed that the coating was evenly distributed on the substrate. The phase identification of the YOF synthesized by laser treatment was evaluated, as shown in Figure 6e. The thin-film-mode measurement revealed a distinct difference before and after the laser coating treatment. Al peaks were observed before the laser treatment (JCPDS 98-000-0062); however, the YOF peaks after the laser treatment were apparent, corresponding to 25.96°, 28.12°, and 40.42° (JCPDS 00-071-2100). The diffraction peaks were attributed to the rhombohedral phase YOF, and no further impurity phases were detected.

XPS analysis was conducted to evaluate the chemical bonding of the synthetically coated materials. Clear peaks were detected for Y and F (Figure 7a). The binding energies of the Y 3d, O 1s, and F 1s orbitals are shown in Figure 7b–d, respectively. The Y $3d_{3/2}$, Y $3d_{5/2}$, and F 1s orbitals indicated 160.5 eV, 158.4 eV, and 685.1 eV, respectively. In particular, Y–OH bonding was observed, implying that YOF was synthesized by a hydrothermal

process in an aqueous solution [42,43], as shown in Figure 7c. The laser-focused region absorbed heat. In addition, it provided a site for hydrothermal synthesis of YOF, because the laser irradiation functioned as a thermal energy source.

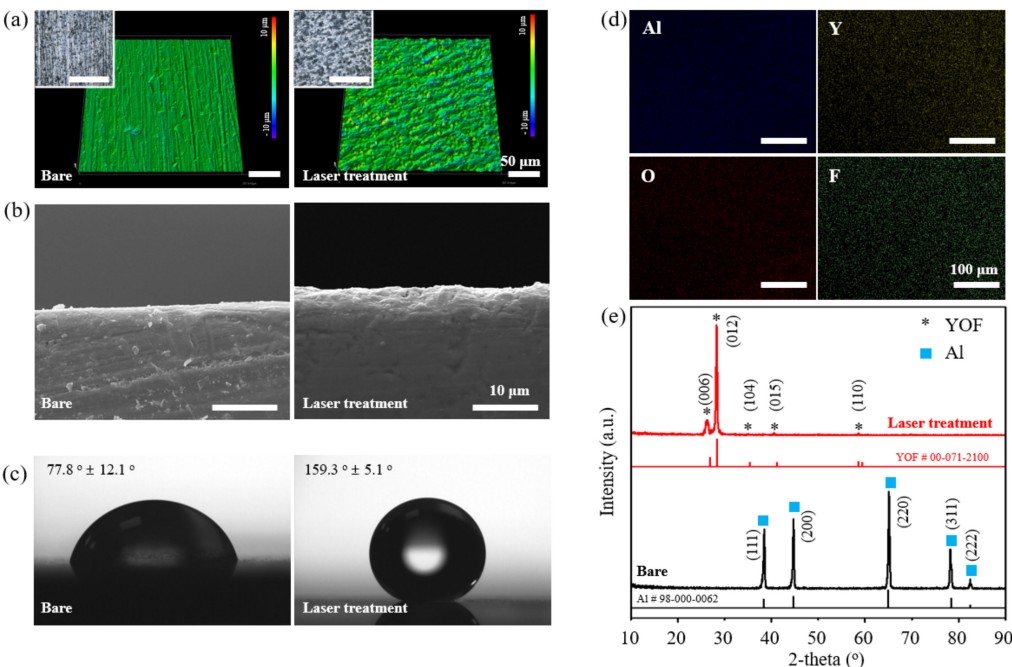

**Figure 6.** Analysis of surface properties after YOF coating via laser treatment: (**a**) Optical (inset) and surface morphology images before and after coating. (**b**) SEM images of the cross-section and (**c**) contact angle of the surface before and after coating. (**d**) Mapping images of Y, O, and F distribution by EDS. (**e**) Phase identification by XRD measurement before and after coating.

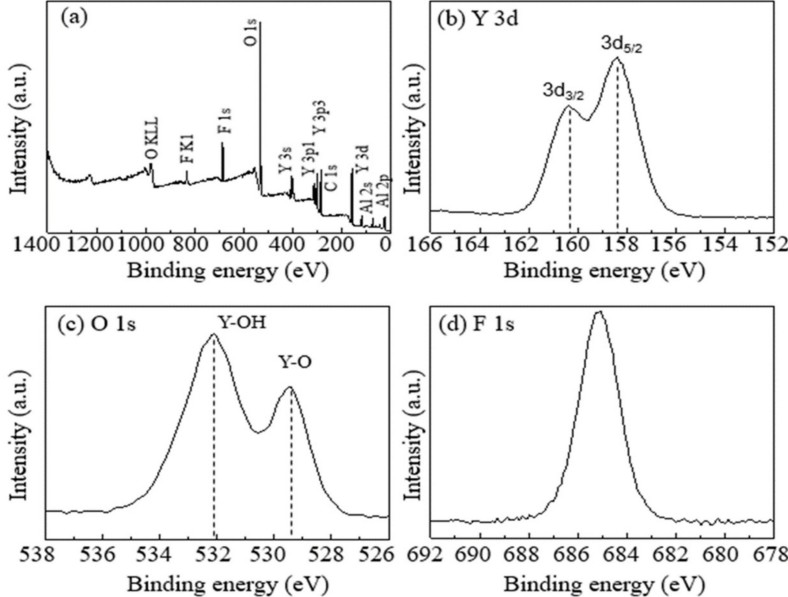

**Figure 7.** X-ray photoelectron spectroscopy (XPS) analysis of the coating layer: (**a**) surface survey, high-resolution of (**b**) Y 3d, (**c**) O 1s, and (**d**) F 1s orbitals.

## 4. Conclusions

YOF was synthesized and coated on an Al substrate by hydrothermal synthesis using LISSC technology in a modified precursor solution. In unmodified yttrium nitrate solutions, there was insufficient time for the synthesis and formation of YOF on the substrate, because

heat energy quickly escaped, owing to the high conductivity of Al. Consequently, YOF was coated firmly on the Al surface by modifying the laser parameters, such as power and loops, as well as the addition of HMTA as a hydroxyl-group-stabilizing chemical. In addition, using X-ray analyses, such as XRD and XPS, it was feasible to demonstrate that the material coated on the surface was YOF by measuring crystallographic and chemical bonding. This shows that precipitation and coating can be achieved by reducing the synthesis time and threshold, by manipulating the laser parameters as well as the precursor composition of the material to be synthesized. This research is expected to be utilized in the field of coating technology, as well as in the synthesis of desired materials.

**Author Contributions:** Drafting the work for the content, analysis, or interpretation of data, J.P.; analysis and interpretation of data for the work, K.L., J.L., H.W.H., G.J. and K.Y.K.; review and analysis of the work, Y.-C.K., M.-R.O., H.-S.H. and J.-Y.S.; conception or design of the work, interpretation of data, and final approval of the manuscript, H.J. All authors have read and agreed to the published version of the manuscript.

**Funding:** This research was supported by the National Research Foundation of Korea (NRF) grant funded by the Korean government (MSIT) (2020R1A2C2010413), KIST project (grant number 2E31641), and KU-KIST Graduate School of Converging Science and Technology Program.

**Institutional Review Board Statement:** Not applicable.

**Informed Consent Statement:** Not applicable.

**Data Availability Statement:** Not applicable.

**Conflicts of Interest:** The authors declare no conflict of interest.

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
