# Peer review of "Improvement of Yttrium Oxyfluoride Coating with Modified Precursor Solution for Laser-Induced Hydrothermal Synthesis"

_coatings, doi:10.3390/coatings12060740_

Round 1
Reviewer 1 Report
The clarity of the manuscript could be improved.
-What is mentioned in Chapter 3.1 under label Y ? Clarify it exactly.
-XRD results are presented "like at some low-priority conference". Add more details and explain.

Reviewer 2 Report
Interesting research results are presented in this manuscript. It is worth considering for publication in this journal as it suits the scope of the journal. However, the SEM results are not explained properly. Besides, there are some minor language corrections which must be addressed. I recommend a major revision for this article.
In the abstract, instead of writing, “… precursor solution modified with chemicals…”, write some words about the solution used in this study.
In section 2.1, please indicate clearly what modifications were made in the precursor solution. In addition to that, please indicate the differences in these modifications with respect to the ones found in literature.
Lines 100 and 101 are not very clearly. Please rewrite them.
In line 113, what does the treated surface represent?
Explanations about the SEM results in Figures 2a, 3a, 4a and 5a-d are not presented very well. Give detailed explanations about the SEM results and their significance to the findings in the research work. If it is possible, please provide a section dedicated to SEM morphologies and explain them in detail.
Round 2
Reviewer 2 Report
The authors have revised their manuscript quite well. I have one last suggestion before accepting this article for publication.
While answering comment #2 from Review Report 1, the authors have given explanations about how the precursor solution was modified. I suppose this statement will be more valid in the article if the authors can supplement a reference from literature to support their argument.
Author Response
We appreciate the valuable comments. As suggested, references that NaF was used to synthesize YOF and HMTA was mixed to form stable oxide were added to the Reference List (Ref 19, 33-35). We consider that the reviewer comment contributed to the improvement of the quality of the paper. Thank you so much for your sincere consideration.
